# Anti-Oxidative and Anti-Aging Effects of Ethanol Extract of the Officinal Breynia (*Breynia vitis-idaea*) In Vitro

**DOI:** 10.3390/plants12051088

**Published:** 2023-03-01

**Authors:** Chae Yun Shin, Jiwon Jang, Hwa Pyoung Lee, Sang Hee Park, Masphal Kry, Omaliss Keo, Byoung-Hee Lee, Wooram Choi, Sarah Lee, Jae Youl Cho

**Affiliations:** 1Department of Biocosmetics, Sungkyunkwan University, Suwon 16419, Republic of Korea; 2Department of Integrative Biotechnology, Sungkyunkwan University, Suwon 16419, Republic of Korea; 3Forestry Administration of the Ministry of Agriculture, Phnom Penh 120206, Cambodia; 4National Institute of Biological Resources, Environmental Research Complex, Incheon 22689, Republic of Korea

**Keywords:** *Breynia vitis-idaea*, anti-aging, anti-oxidative

## Abstract

The skin is the largest organ of the human body, and it is also the one most exposed to external environmental contaminants. The skin is the body’s first defense against harmful environmental stimuli, including ultraviolet B (UVB) rays and hazardous chemicals. Therefore, proper care of the skin is required to prevent skin-related diseases and age-related symptoms. In this study, we analyzed anti-aging and anti-oxidative effects of *Breynia vitis-idaea* ethanol extract (Bv-EE) in human keratinocytes and dermal fibroblasts. The Bv-EE had free radical scavenging activity and decreased the mRNA expression of *MMPs* and *COX-2* in H_2_O_2_- or UVB-treated HaCaT cells. The Bv-EE also inhibited AP-1 transcriptional activity and phosphorylation of c-Jun N-terminal kinase, extracellular signal-regulated kinase, and mitogen-activated protein kinase 14 (p38), which are major AP-1 activators upon H_2_O_2_ or UVB exposure. Furthermore, the promoter activity and mRNA expression of collagen type I (Col1A1) increased in HDF cells treated with Bv-EE, and Bv-EE recovered the collagen mRNA expression decreased by H_2_O_2_ or UVB exposure. These results suggest that Bv-EE has anti-oxidative effects by inhibiting the AP-1 signaling pathway, and shows anti-aging effects by upregulating collagen synthesis.

## 1. Introduction

The genus *Breynia* belongs to the family Phyllanthaceae, which is composed of 35 plant species distributed in tropical regions of the Pacific Islands, Australia, and Asia [1]. Many reports about the medicinal benefits of *Breynia* species have been published. *Breynia nivosa* possesses analgesic, anti-inflammatory, and antimicrobial properties [2]. *Breynia retusa* shows antioxidant and anti-diabetic activities, and *Breynia distachia* has shown hypoglycemic and anti-Alzheimer’s activities [3,4]. *Breynia vitis-idaea*, commonly called Indian snowberry, is a large, evergreen shrub or treelet that grows up to 5 m in height [5]. These plants have ovate or elliptic leaves that are 1–3 cm long. The bark is yellowish grey, the flowers are small and greenish yellow or pink, and the fruits are dull red, purple, or white berries [6]. These plants are found in countries including Bangladesh, Cambodia, India, Malaya, Myanmar, Pakistan, Philippines, Sri Lanka, Thailand, and Vietnam, and have been used as a traditional herbal medicine for the treatment of wounds and chronic bronchitis, especially in China [7]. Recent studies have revealed that *Breynia vitis-idaea* has anti-hypoglycemic, anti-hypolipidemic, and anti-cancer activity [8,9]. Interestingly, *Breynia vitis-idaea* is known to contain 6-O-benzoyl arbutin, breynioside B, and 6-O-benzoyl-α-D-glucose, which are known to have antioxidant activity [5]. Despite the various therapeutic potentials of *Breynia vitis-idaea*, its ability to inhibit skin aging has not been studied. 

The skin is the largest organ of the human body and acts as the first protective system from external threats such as noxious substances and pathogens [10,11]. Aging weakens the skin’s protective function, resulting in slow wound repair, frequent skin inflammation, and increased risk of skin cancer [12,13,14]. Skin aging is a cumulative process involving intrinsic and extrinsic factors. Among the extrinsic factors, photoaging constitutes as much as 80% of skin aging [15]. Repetitive exposure of skin to UV radiation causes several skin problems, including loss of elasticity, deformation, sagging, and wrinkling [16]. UV radiation is classified as UVA (320–400 nm), UVB (290–320 nm), and UVC (200–290 nm). Although most UVC radiation is absorbed by the atmosphere, UVA and UVB rays can directly affect the skin, and the high-energy, short-wavelength UVB is more threatening to the skin than UVA [17]. Exposure of skin cells to UVB radiation triggers a signaling response called the UV response, which includes activation of the mitogen-activated protein kinase (MAPK) cascade [18]. MAPK activation induces the expression of cyclooxygenase-2 (*COX-2*), which catalyzes the conversion of arachidonic acid into prostaglandins and is closely implicated in aging processes [19,20,21]. UVB-induced activation of MAPK is followed by enhanced transcriptional activity of the activator protein-1 (AP-1) complex, which consists of heterodimers of c-Jun and c-Fos proteins [22,23]. Activation of the AP-1 complex leads to upregulation of extracellular-matrix (ECM)-degrading enzymes, such as matrix metalloproteinase-3 (*MMP-3*) and *MMP-9*, which are responsible for degrading the collagen and ECM that compose the dermal connective tissue [24]. Protecting the skin from UVB exposure is one way to delay skin aging. Therefore, our aim in this study was to investigate the anti-aging and antioxidant effects of an ethanol extract of *Breynia vitis-idaea* (Bv-EE) in UVB- and H_2_O_2_-damaged human skin cell lines.

## 2. Results

### 2.1. Phytochemical Components of Bv-EE

The phytochemical composition of Bv-EE was analyzed by gas chromatography–mass spectrometry (GC–MS) (Figure 1). The corrected percentage peak area was divided by the sum of the corrected area to obtain the total content of each compound in Bv-EE. The major compound was identified as 2,3-dihydro-3,5-dihydroxy-6-methyl- 4H-pyran-4-one, also known as DDMP. This compound is produced in the intermediate stage of the Maillard reaction [25]. The contribution of DDMP to antioxidant activity has been well studied, and DDMP is known to exist in natural extracts and foods [26,27]. In particular, DDMP is known for its antioxidant effect and its ability to prevent and treat obesity or lipid-related metabolic diseases [28]. Other active components of Bv-EE are listed in Table 1. Nonetheless, confirming the identification of these components from Bv-EE will be continued with other extracts prepared with the same plant materials harvested according to different seasons, regions, and years to complete standardization of this extract for industrial application.

### 2.2. Bv-EE Exerts Reactive Oxygen Species Scavenging Activity

To determine whether Bv-EE has radical scavenging activity, 1-diphenyl-2-picryl-dydrazyl (DPPH) and 2,2’-azino-bis (3-ethylbenzothiazoline-6-sulphonic acid) diammonium salt (ABTS) assays were performed [29]. The Bv-EE scavenged DPPH radicals in a dose-dependent manner and showed significant scavenging activity at a concentration of 6.25 μg/mL (Figure 2a). In the ABTS assay (Figure 2b), Bv-EE exhibited ABTS radical scavenging activity, with IC_50_ values of 10.8424 μg/mL and 5.36 μg/mL (Figure 2c,d). Ascorbic acid (300 μM) was used as a positive control (Figure 2a,b). The ferric reducing antioxidant power (FRAP) assay showed that Bv-EE can also reduce ferric acid dose-dependently (Figure 2e). In addition, the cupric ion-reducing antioxidant capacity (CUPRAC) assay showed that Bv-EE reduced Cu ions in a dose-dependent manner. Trolox (3 mg/mL) was used as the positive control. Moreover, we determined the total phenolic content and total flavonoid content of Bv-EE to be 89.33 μg/mg and 73 μg/mg, respectively (Figure 2g,h). These results together show that Bv-EE is a potential antioxidant.

### 2.3. Bv-EE Prevents Cell Death Caused by UVB and H_2_O_2_ in HaCaT cells

Because Bv-EE showed ROS scavenging activity, we tested whether it has the same effect in human keratinocytes. First, we determined whether Bv-EE has cytotoxic effects on HaCaT cells. As shown in Figure 3a, Bv-EE did not show cytotoxicity and did not impact cell viability. Next, we induced cell death using UVB (30 mJ/cm^2^) or H_2_O_2_ (200 μM) and found that the decreased cell number recovered with Bv-EE treatment (Figure 3b,c). In addition, to investigate whether Bv-EE can reduce ROS generation in keratinocytes, we stained HaCaT cells with 6-diamidino-2-phenylindole (DAPI) and 2′,7′–dichlorodihydrofluorescein diacetate (H2DCFDA) after Bv-EE treatment. The ROS level increased upon treatment with H_2_O_2_; however, Bv-EE treatment reduced the ROS level in a dose-dependent manner. We also analyzed the ROS level in HaCaT cells using flow cytometry and similarly found that ROS generation decreased upon Bv-EE treatment (Figure 3d). Thus, Bv-EE can reduce the ROS generation caused by UVB exposure or H_2_O_2_ treatment, suggesting that it has antioxidant and anti-aging effects.

### 2.4. Bv-EE Inhibits AP-1 Promoter Activity and mRNA Expression of Aging Factors

Because Bv-EE was able to inhibit intracellular ROS generation and increase cell viability, we examined whether those effects involve AP-1, which is a main transcription factor that regulates ROS generation and cellular responses [30]. After confirming that Bv-EE has no cytotoxic effect on HEK293T cells (Figure 4a), we examined AP-1 promoter activity using a luciferase assay. As shown in Figure 4b, the AP-1 promoter activation level decreased significantly in a dose-dependent manner upon Bv-EE treatment in Toll-interleukin-1 receptor-domain-containing adapter-inducing interferon-β (TRIF)-induced HEK293T cells. Because Bv-EE can suppress AP-1 transcriptional activity, and AP-1 is a major regulator of the transcription of aging- and oxidative-related factors [31], we analyzed whether the mRNA levels of those related factors were regulated by Bv-EE. Interestingly, the mRNA expression of *MMP-3* and *MMP-9* was decreased by Bv-EE in H_2_O_2_-exposed HaCaT cells (Figure 4c). The *COX-2* expression level was also decreased in the Bv-EE-treated groups in a dose-dependent manner (Figure 4d). In addition, we investigated the expression of those factors in UVB-treated HaCaT cells. The mRNA expression of *MMP-3* and *MMP-9* was reduced upon Bv-EE treatment (Figure 4e), and the *COX-2* expression level was inhibited dose-dependently (Figure 4f). Collectively, these data indicate that key factors related to aging and oxidation were inhibited by Bv-EE as a consequence of its inhibition of AP-1.

### 2.5. Bv-EE Inhibits the AP-1 Signaling Pathway

We hypothesized that Bv-EE inhibited AP-1 transcription by inhibiting an upstream signaling pathway. Therefore, we studied whether phosphorylation of the kinases involved in the AP-1 signaling pathway would be reduced by Bv-EE in UVB- or H_2_O_2_-treated HaCaT cells. First, we measured the phosphorylation levels of c-Jun and c-Fos, which are two main subunits of AP-1 [32]. Upon UVB irradiation, the expression of phosphorylated c-Jun and c-Fos was dramatically increased, and Bv-EE treatment decreased that expression in a dose-dependent manner (Figure 5a–c). We also examined the phosphorylation of AP-1 upstream kinases. Increased phosphorylation of the c-Jun N-terminal kinase (JNK) and p38 was decreased by Bv-EE treatment (Figure 5d–f). We then examined the same factors in cells subjected to H_2_O_2_ instead of UVB. We found downregulated phosphorylation of JNK, extracellular signal-regulated kinase (ERK), and p38, though the total form was unchanged (Figure 5g–i). These results indicate that Bv-EE can inhibit phosphorylation of the AP-1 signaling pathway to protect cells from UVB irradiation and oxidative stress.

### 2.6. Bv-EE Promotes Collagen Generation in Human Dermal Fibroblast (HDF) Cells

Aging and oxidative stress play important roles in wrinkle formation. Therefore, wrinkles are frequently considered an indicator of aging and oxidation [33,34]. Based on that concept, we tested whether Bv-EE can enhance the expression of wrinkle-related factors in human dermal fibroblast cells, which are responsible for collagen generation. After confirming that Bv-EE has no cytotoxic effect on HDF cells (Figure 6a), we examined whether Bv-EE can regulate Col1A1, a typical collagen-encoding gene [35]. As shown in Figure 6b, Bv-EE treatment dose-dependently upregulated Col1A1 promoter activity; retinol was used as a positive control. Correspondingly, mRNA expression of Col1A1 was also increased by Bv-EE treatment (Figure 6c). In addition, we tested whether Bv-EE can restore the Col1A1 expression decreased by exposure to UVB or H_2_O_2_. Interestingly, we found that the decreased mRNA expression of Col1A1 was recovered by Bv-EE treatment (Figure 6d,e). These results indicate that Bv-EE can promote collagen synthesis by upregulating its transcriptional activity.

## 3. Discussion

Organic and nature-derived materials are attractive approaches to skin therapy due to their minimal toxicity and side effects [36]. This is an important factor in pharmaceutical and cosmetic formulations. In this study, we used UVB- and H_2_O_2_-induced damage in vitro to study the efficacy of a novel plant extract, Bv-EE, and demonstrated its protective effects in human keratinocytes and dermal fibroblasts. Importantly, Bv-EE showed no cytotoxicity to HaCaT or HDF cells, which are the cell lines most widely used for human skin research.

Oxidative stress is the largest cause of skin damage and is associated with skin aging [37]. Therefore, the use of antioxidants has become a leading approach for anti-aging therapy [38]. Several chemicals have been approved for application in the pharmaceutical and cosmetic industries [39]. In this work, we found that Bv-EE has ROS scavenging activity (Figure 2), indicating that it can directly scavenge generated ROS. More importantly, Bv-EE inhibits ROS generation in human keratinocytes (Figure 3). UVB and H_2_O_2_ treatment induce serious oxidation processes in cells and eventually lead to cell death. Bv-EE showed a remarkable ability to prevent cell death caused by ROS responses, indicating the potential of Bv-EE as a drug to treat skin diseases.

When *COX-2* and *MMPs* are produced in response to UVB irradiation, they play an important role in inflammatory responses in skin cells. *COX-2* mediates inflammatory processes in the skin, including inflammatory hyperalgesia and nociception [40,41,42]. In this work, we discovered that Bv-EE treatment significantly decreased *COX-2* and *MMP* transcriptional activity and mRNA expression in UVB- and H_2_O_2_-damaged keratinocytes (Figure 4). That finding clearly indicates that Bv-EE acts as an antioxidant, as well as an anti-aging agent, by recovering the damage induced by UVB irradiation and oxidative stress.

The AP-1 signaling pathway is prominently activated by external triggers, including UVB and oxidative stress. Therefore, its constitutive kinases are promising targets for disease therapies [43,44]. In this study, we focused on MAPK-related enzymes and confirmed that Bv-EE can inhibit the phosphorylation of JNK, ERK, and p38 in UVB- and H_2_O_2_-damaged keratinocytes (Figure 5). These findings suggest that the skin-protective characteristics of Bv-EE occur by regulating the AP-1 intracellular molecular signaling pathway.

One of the most typical symptoms and indicators of aging is wrinkles, which are caused by a decrease in ECM proteins such as collagen in fibroblast cells [45]. Therefore, the proper production of collagen is an important process for maintaining healthy skin [46]. In this work, we found that Bv-EE promotes collagen synthesis by increasing Col1A1 transcriptional activity and mRNA expression in human dermal fibroblasts (Figure 6). More interestingly, Bv-EE can restore the collagen synthesis decreased by UVB irradiation and H_2_O_2_ treatment (Figure 6). These findings strongly suggest that Bv-EE can help to prevent skin wrinkles by promoting collagen synthesis.

Among all the skin-protecting agents currently used in the pharmaceutical and cosmetic industries, Bv-EE has shown its potential as a natural compound with anti-aging and antioxidant properties. This study provides novel insights about an organic and natural ingredient as a promising drug candidate because of its genetic and molecular regulation of potential targets.

## 4. Materials and Methods

### 4.1. Materials

HaCaT, HDF, and HEK293T cells were purchased from the American Type Culture Collection (Rockville, MD, USA). Dulbecco’s modified Eagle’s medium (DMEM), fetal bovine serum (FBS), phosphate-buffered saline (PBS), penicillin–streptomycin, bovine serum albumin (BSA), gallic acid, quercetin, aluminum chloride, DPPH, ABTS, potassium sulfate, trypsin, and ascorbic acid were purchased from Hyclone (Grand Island, NY, USA). Retinol, dimethyl sulfoxide (DMSO), 3-(4,5-dimethylthiazol-2-yl)-2,5-diphenyl tetrazolium bromide (MTT), polyethylenimine (PEI), TRIzol, hydrogen peroxide (H_2_O_2_), 2,4,6-tri(2-pyridyl)-s-triazine (TPTZ), FeCl_3_·6H_2_O, Trolox, CuCl_2_·2H_2_O, NH4Ac, neocuproine, 3-(4,5-dimethylthiazol-2′, 7′-dichlorofluorescein diacetate (DCFDA), and DAPI were purchased from Sigma Aldrich Chemical Co. (St. Louis, MO, USA). The MTT stop solution was prepared by adding 10% sodium dodecyl sulfate to hydrochloric acid (HCl). The luciferase assay system kit was obtained from Promega (Madison, WI, USA). The cDNA synthesis kit was received from Thermo Fisher Scientific (Waltham, MA, USA). The forward and reverse primers used in RT-PCR were synthesized by Macrogen (Seoul, Republic of Korea), and the PCR premix was obtained from Bio-D Inc. (Seoul, Republic of Korea). The polyvinylidene difluoride (PVDF) membranes and enhanced chemiluminescence (ECL) reagent were purchased from Bio-Rad (Hercules, CA, USA). Specific antibodies for the total- and phosphorylated forms of ERK, JNK, p38, c-Jun, c-Fos, and β-actin were purchased from either Cell Signaling Technology (Beverly, MA, USA) or Santa Cruz Biotechnology (Santa Cruz, CA, USA).

### 4.2. Preparation of Breynia vitis-idaea

*Breynia vitis-idaea* was procured from the National Institute for Biological Resources (Incheon, Republic of Korea). The dried stems (1.3 kg) were extracted in 70% ethanol (3 × 18 L) at room temperature for 3 h using an ultrasonicator (Ultrasonic Cleaner UC-10, UC-20, 400 W) under dark conditions. After removing the macerate by filtration, the extracted solution was concentrated in vacuo at 40 °C using a rotary evaporator (IVT Co., Ltd., Daegu, Republic of Korea), and then freeze-dried for 48 h at −80 °C to make a completely dried powder (color of the extract: brown). The yield of this extraction was 13.97%.

### 4.3. GC–MS

The GC–MS analysis of dried Bv-EE (100 mg/mL in methanol) was carried out with an Agilent 8890 GC instrument (Santa Clara, CA, USA) equipped with an Agilent J&W DB-624 Ultra Insert GC column (60 m in length × 250 µm in diameter × 1.40 µm in thickness), and mass spectrometry was conducted with an Agilent 5977B MSD instrument (Santa Clara, CA, USA) equipped with a Series II triple-axis detector with a high energy dynode and long-life electron multiplier from the Cooperative Center for Research facilities of Sungkyunkwan University (Suwon, Republic of Korea), as previously reported [47,48]. Detailed conditions of the analysis are listed in Table 2. The spectrum of phytochemicals in the National Institute of Standards and Technology library was used to identify the unknown phytochemicals in Ca-EE, as reported previously [49,50].

### 4.4. Cell Culture

HaCaT cells (a human keratinocyte cell line) and HDF cells (a human fibroblast cell line) were cultured in DMEM supplemented with 10% FBS and 1% penicillin–streptomycin, and HEK293T cells (a human embryonic kidney cell line) were cultured in DMEM supplemented with 5% FBS and 1% penicillin–streptomycin. All cells were maintained in a 5% CO_2_ incubator at 37 °C.

### 4.5. DPPH Radical Scavenging Activity Assay

The DPPH assay was performed to determine the radical scavenging capacity of the extract. We dissolved 300 μM DPPH in methanol and used ascorbic acid (100 μM) dissolved in PBS as the positive control. Different concentrations of Bv-EE were prepared, and the absorbance was measured at 517 nm and 37 °C for 15 min. Radical scavenging activity was calculated using the following equation, as reported previously [51,52].
DPPH radical scavenging activity % = [(A_0_ − A_1_)/A_0_] × 100
where A_0_ is the absorbance of DPPH, and A_1_ is the absorbance of the extract.

### 4.6. ABTS Radical Scavenging Activity Assay

The ABTS assay was performed to determine the radical scavenging capacity of Bv-EE. First, 7.4 mM ABTS dissolved in PBS and 2.4 mM of potassium persulfate dissolved in PBS were mixed at a ratio of 1:1 and incubated overnight. Different concentrations of Bv-EE and ascorbic acid (100 μM) were prepared, and the absorbance was measured at 730 nm and 37 °C for 15 min. The percentage of scavenging was calculated using the same method as stated previously [53].

### 4.7. CUPRAC Assay

The CUPRAC assay was performed to determine the cupric reducing antioxidant capacity of Bv-EE. First, 100 mM CuCl_2_·2H_2_O (copper (II) chloride solution) was dissolved in distilled or deionized water. NH_4_Ac (ammonium acetate) was dissolved in distilled or deionized water, and the pH was adjusted to 7.0. Neocuproine (Nc) solution (7.5 mM) was dissolved in pure ethanol. Next, the copper (II) chloride solution, ammonium acetate buffer, and Nc solution were mixed at a ratio of 1:1:1 to a final volume of 200 μL in e-tubes, and 200 μL of Bv-EE solution was added to each tube. Trolox (3 mg/mL) dissolved in pure ethanol was used as a positive control. After a 1 h incubation at room temperature, the mixed solution was transferred to a 96-well plate, and the absorbance was measured at 450 nm [54].

### 4.8. FRAP Assay

The FRAP assay was performed to determine the ferric reducing power of Bv-EE. First, 300 mM acetic acid buffer was prepared and mixed with anhydrous sodium acetate and glacial acetic acid (pH 3.6). Next, 10 mM TPTZ solution dissolved in distilled or deionized water and 20 mM FeCl_3_·6H_2_O dissolved in distilled or deionized water were mixed with the FeCl_3_ solution at a ratio of 10:1:1. Bv-EE solution was prepared at different concentrations and aliquoted to a 96-well plate at 100 μL per well. Then, 100 μL of FRAP working solution were added to each well and incubated for 15 min at 37 °C in the dark. Trolox was used as a positive control, and absorbance was measured at 593 nm [55].

### 4.9. Determination of Total Phenolic Content

At room temperature, 100 μL of Bv-EE (serially diluted five times in distilled or deionized water) and 100 μL of 10% Folin–Ciocâlteu reagent were mixed with 300 μL of distilled or deionized water and incubated for 5 min. Next, 500 μL of 8% sodium carbonate and 500 μL of distilled or deionized water were added to the tubes. After 30 min of incubation at room temperature in a dark room, the mixture was transferred to a 96-well plate, and the absorbance was measured at 765 nm. The calculation was performed using a standard curve obtained with gallic acid, and the total phenolic content is expressed as mg of gallic acid equivalent/g of extract.

### 4.10. Determination of Total Flavonoid Content

For this analysis, 100 μL of Bv-EE (serially diluted five times in distilled or deionized water) and 100 μL of aluminum chloride 2% reagent were mixed in a 1:1 ratio, as reported previously [56,57]. After a 1 h incubation at room temperature in a dark room, the mixture was transferred to a 96-well plate, and the absorbance was measured at 420 nm. The calculation was performed using a standard curve obtained with quercetin, and the total flavonoid content is expressed as mg of quercetin equivalent/g of extract.

### 4.11. ROS Generation Assay

The 2′,7′–dichlorodihydrofluorescin diacetate (H2DCFDA) assay was used to detect intracellular ROS. HaCaT cells were pretreated with Bv-EE for 30 min and then treated with H_2_O_2_ (200 μM) for 24 h. The cells were washed with PBS, stained with 10 μM H2DCFDA, and incubated for 20 min in the dark. The cells were then fixed in formaldehyde solution (100 μL/mL) for 10 min, washed with PBS two times, stained with DAPI (1 μL/mL), and incubated for 20 min in the dark. Photographs were captured using a Nikon Eclipse Ti (Nikon, Japan) fluorescence microscope.

For flow cytometry, HaCaT cells were treated with Bv-EE and exposed to H_2_O_2_, as stated above. The cells were harvested and resuspended in 300 μL of PBS. Then, 10 μM H2DCFDA was added to the tube and incubated for 20 min in the dark. The fluorescence was detected at 485/535 nm using a flow cytometer (Beckman Coulter, Brea, CA, USA), as described previously [58].

### 4.12. Cell Viability Assay

HaCaT and HEK293T cells were seeded at 3 × 10^5^ cells/mL, and HDF cells were seeded at 1 × 10^5^ cells/mL in 96-well plates and incubated for 24 h. Then, Bv-EE was added to all cells for 24 h. Next, 100 μL of the original media was removed, and 10 μL of MTT solution was added to each well and incubated for 4 h. To each well, 100 μL of MTT stopping solution was added and incubated overnight. The absorbance was measured at 570 nm using a multi-plate microreader, as previously reported [59,60].

### 4.13. H_2_O_2_ Treatment and UVB Irradiation

HaCaT cells were plated in 6-well plates at 2 × 10^5^ cells/mL and incubated for 24 h. The cells were pretreated with Bv-EE for 30 min, washed with cold PBS, and exposed to H_2_O_2_ (200 μM) or UVB radiation (30 mJ/cm^2^). After that, the cells were treated with Bv-EE at different concentrations for 24 h, as previously reported [61]. A BLX-312 (Vilber Lourmat, France) UVB lamp was used for UVB irradiation. Cell viability was calculated as follows:Cell viability (% of control) = A_1_/A_0_ × 100
where A_1_ refers to treated cells, and A_0_ refers to normal untreated cells.

### 4.14. Semi-Quantitative Reverse Transcription-PCR (RT-PCR) and Quantitative Real-Time PCR (Real-Time PCR)

HaCaT and HDF cells were exposed to UVB and H_2_O_2_ as described above. Either Bv-EE or retinol (10 μg/mL) was added to the cells for 24 h. RNA was isolated using TRI reagent solution. cDNA was synthesized from total RNA (1 μg) using a cDNA synthesis kit (Thermo Fisher Scientific, Waltham, MA, USA). RT-PCR and real-time PCR were conducted as previously described [62,63]. The primer sequences used in this experiment are listed in Table 3.

### 4.15. Luciferase Reporter Gene Assay

HEK293T cells were seeded in a 24-well plate at 3 × 10^5^ cells/mL. After 18 h of incubation, the HEK293T cells were co-transfected with luciferase-expressing genes (AP-1 and Col1A1) and the β-galactosidase gene using PEI. After 24 h of incubation, the cells were treated with Bv-EE or retinol for 24 h. Then, 300 μL of luciferase lysis buffer was added to each well, and the plate was frozen for 3 h at −70 °C. The luciferase assay was conducted using a luciferase assay system reported previously [64].

### 4.16. Preparation of Whole Cell Lysates and Western Blot Analysis

HaCaT cells were washed with cold PBS, collected using a cell scraper, and centrifuged at 12,000 rpm for 5 min at 4 °C. The cells were lysed for 15 min on ice in cell lysis buffer (20 mM Tris-HCl, pH 7.4; 2 mM EDTA; 2 mM ethyleneglycotetraacetic acid; 1 mM dithiothreitol; 50 mM β-glycerol phosphate; 0.1 mM sodium vanadate; 1.6 mM pervanadate; 1% Triton X-100; 10% glycerol; 10 μg/mL aprotinin; 10 μg/mL pepstatin; 1 µM benzamide; and 2 μM phenylmethylsulfonyl fluoride) and kept at −70 °C until use. The supernatant containing protein was collected and used for Western blotting. Protein concentrations were measured using the Bradford assay as described previously [65]. For that, 20 μg of protein from each sample was separated by 10% sodium dodecyl sulfate-polyacrylamide gel electrophoresis and then transferred onto PVDF membranes (Millipore, Billerica, MA, USA). After blocking the membranes with 3% BSA for 1 h at room temperature, we washed the membranes with tris-buffered saline (50 mM Tris-Cl, pH 7.5, 150 mM NaCl) and 0.1% Tween-20 (TBST) three times at 10 min intervals. The membranes were incubated with primary antibodies (1:2500 dilution) overnight at 4 °C. Then, the membranes were washed with TBST three times for 10 min each and incubated with secondary antibody for 2 h at room temperature. After washing the membranes with TBST three times for 10 min each time, we detected their chemiluminescence using ECL reagent [66]. Relative band intensities were measured using ImageJ software (Wayne Rasband, NIH, Bethesda, MD, USA).

### 4.17. Statistical Analyses

All data are presented as the mean ± standard deviation of at least three independent experiments. A Mann–Whitney test was used to compare statistical differences between experimental and control groups. *p*-values < 0.05 were considered statistically significant. All statistical analyses were conducted using SPSS (SPSS, Chicago, IL, USA). 

## 5. Conclusions

Herein, we demonstrated the anti-aging and antioxidant capacity of Bv-EE by evaluating its inhibitory effect on ROS in human keratinocytes and dermal fibroblasts exposed to UVB irradiation and H_2_O_2_. Bv-EE was able to downregulate aging factors (*COX-2* and *MMPs*). Correspondingly, the AP-1 (c-Jun/c-Fos) signaling activity was reduced upon Bv-EE treatment, as summarized in Figure 7. On the other hand, Bv-EE upregulated *Col1A1* expression in human dermal fibroblasts. 

## Figures and Tables

**Figure 1 plants-12-01088-f001:**
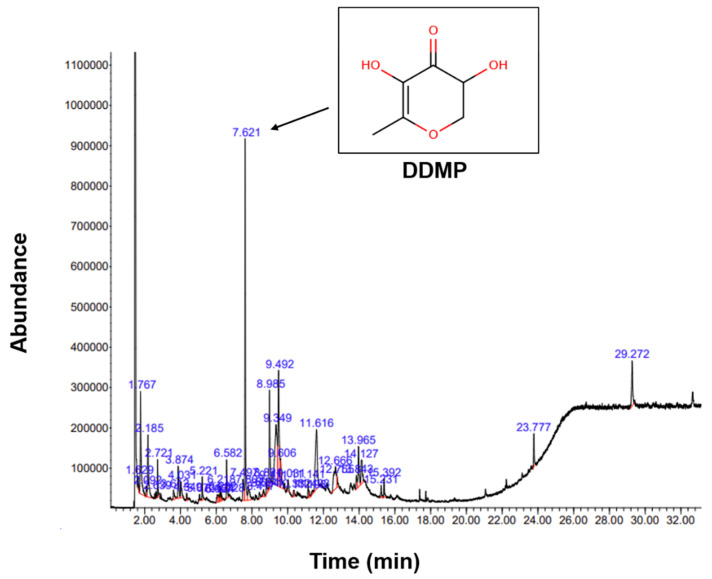
The GC–MS chromatogram of Bv-EE.

**Figure 2 plants-12-01088-f002:**
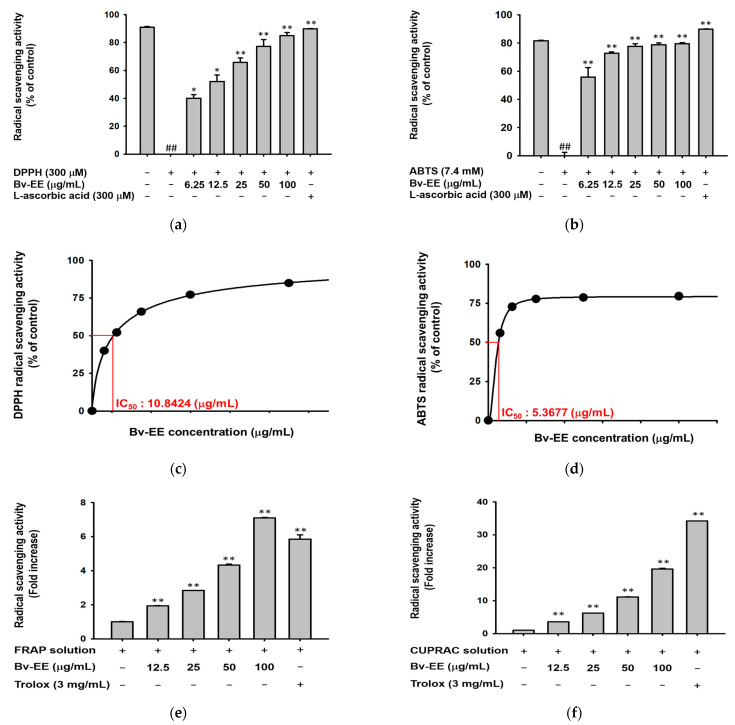
Bv-EE exerts ROS scavenging activity. The radial scavenging activity of Bv-EE was evaluated using the (**a**) DPPH and (**b**) ABTS assays. Ascorbic acid (300 μM) was used as a positive control. (**c**,**d**) The DPPH and ABTS IC_50_ values of Bv-EE. The ability of Bv-EE to use the metal reduction principle was shown in the (**e**) FRAP and (**f**) CUPRAC assays. Trolox (3 mg/mL) was used as a positive control. (**g**) The total phenolic content of Bv-EE expressed as mg of gallic acid equivalent/g of extract. (**h**) The total flavonoid content of Bv-EE expressed as mg of quercetin equivalent/g extract. All results are expressed as means ± standard deviations. ## *p* < 0.01 compared with the positive control group, and * *p* < 0.05, ** *p* < 0.01 compared with the normal group.

**Figure 3 plants-12-01088-f003:**
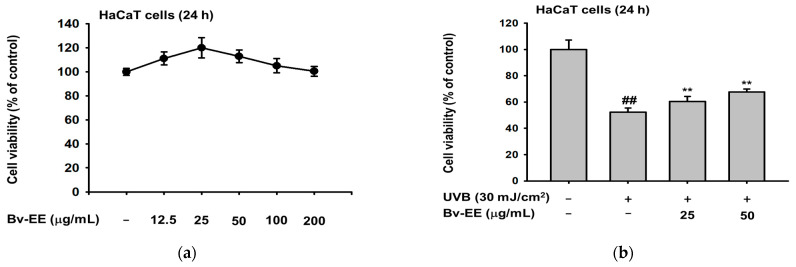
The Bv-EE prevented cell death caused by UVB and H_2_O_2_ in HaCaT cells. (**a**) HaCaT cells were treated with different doses of Bv-EE (12.5–200 μg/mL) for 24 h, and viability was measured using the MTT cell viability assay. (**b**) After UVB exposure (30 mJ/cm^2^), the cells were treated with Bv-EE for 24 h, and viability was measured using the MTT assay. (**c**) HaCaT cells were pretreated with Bv-EE for 30 min and then subjected to H_2_O_2_ (200 μM) for 24 h. (**d**) The ROS level in HaCaT cells was detected using a fluorescence microscope and flow cytometry. (**e**) The relative intensity of the ROS level was determined using ImageJ. All results are expressed as means ± standard deviations. ## *p* < 0.01 compared with the positive control group, and * *p* < 0.05, ** *p* < 0.01 compared with the normal group.

**Figure 4 plants-12-01088-f004:**
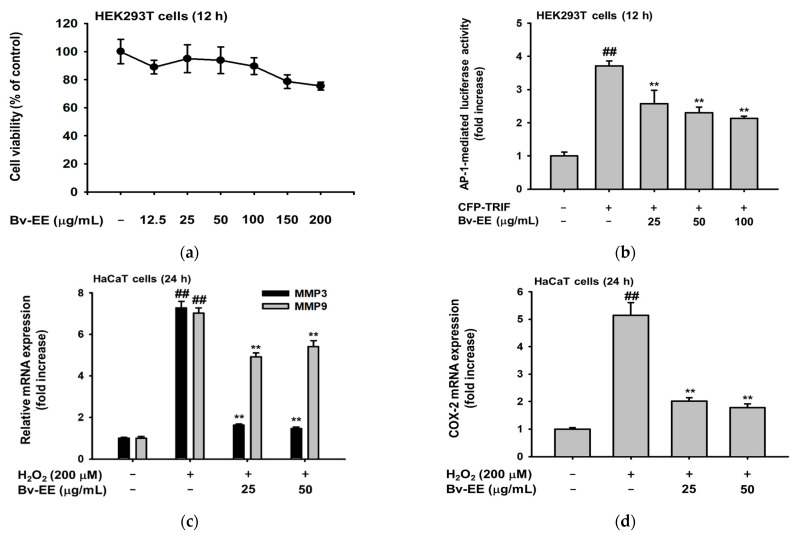
Bv-EE inhibits AP-1 promoter activity and mRNA expression of aging factors. (**a**) HEK293T cells were treated with different concentrations of Bv-EE for 24 h, and viability was measured using the MTT assay. (**b**) HEK293T cells transfected with AP-1-Luc, TRIF, β-galactosidase (control), and PEI were incubated for 24 h and examined by luminometer. (**c**–**f**) Real time PCR analyses of the expression of *MMP3, MMP9,* and *COX-2* in HaCaT cells pretreated with Bv-EE and then subjected to H_2_O_2_ (200 μM) or UVB (30 mJ/cm^2^) for 24 h. The fold increase represents the ratio between the increase in mRNA expression in the Bv-EE treated groups and that in the normal group. GAPDH was used as an internal control. All results are expressed as means ± standard deviations. # *p* < 0.05, ## *p* < 0.01 compared with the positive control group, and ** *p* < 0.01 compared with the normal group.

**Figure 5 plants-12-01088-f005:**
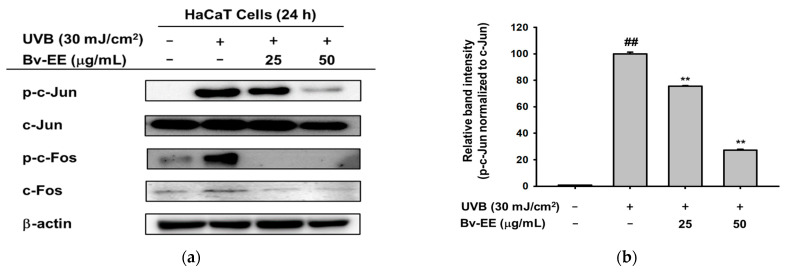
Bv-EE inhibits the AP-1 signaling pathway. (**a**) HaCaT cells were incubated with Bv-EE and irradiated with UVB for 24 h. Then, the phosphorylated protein levels of c-Jun and c-Fos were examined by Western blot. (**d**,**g**) The phosphorylation of ERK, JNK, and p38 was determined by Western blotting following Bv-EE treatment and exposure to UVB or H_2_O_2_. β-actin was used as an internal control (**b**,**c**,**e**,**f**,**h**,**i**) Band intensity was measured and quantified using ImageJ. All results are expressed as means ± standard deviations. ## *p* < 0.01 compared with the positive control group, and ** *p* < 0.01 compared with the normal group.

**Figure 6 plants-12-01088-f006:**
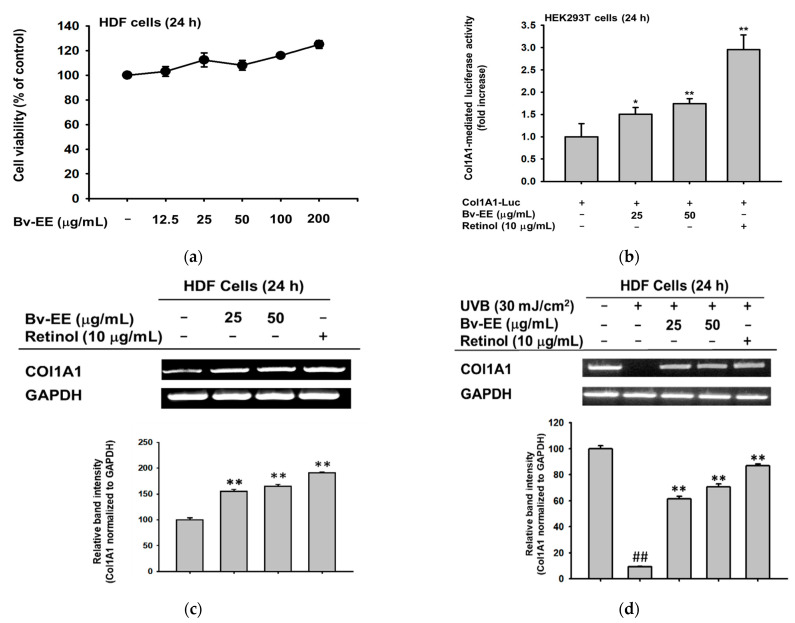
Bv-EE promotes collagen generation in human dermal fibroblast cells. (**a**) The viability of HDF cells treated with Bv-EE (12.5–200 μg/mL) was measured using the MTT assay. (**b**) HEK293T cells transfected with Col1A1-Luc were treated with Bv-EE (25 and 50 μg/mL) or retinol (10 μg/mL) for 24 h and then examined by luminometer. (**c**) HDF cells were treated with different concentrations of Bv-EE (25 and 50 μg/mL) or retinol (10 μg/mL) for 24 h. (**d**) HDF cells were pretreated with Bv-EE for 30 min and then subjected to H_2_O_2_ for 24 h. (**e**) HDF cells were irradiated with UVB and treated with Bv-EE for 24 h. (**c**–**e**) The mRNA expression level of *Col1A1* was determined using semiquantitative RT-PCR. All results are expressed as means ± standard deviations. ## *p* < 0.01 compared with the positive control group, and * *p* < 0.05, ** *p* < 0.01 compared with the normal group.

**Figure 7 plants-12-01088-f007:**
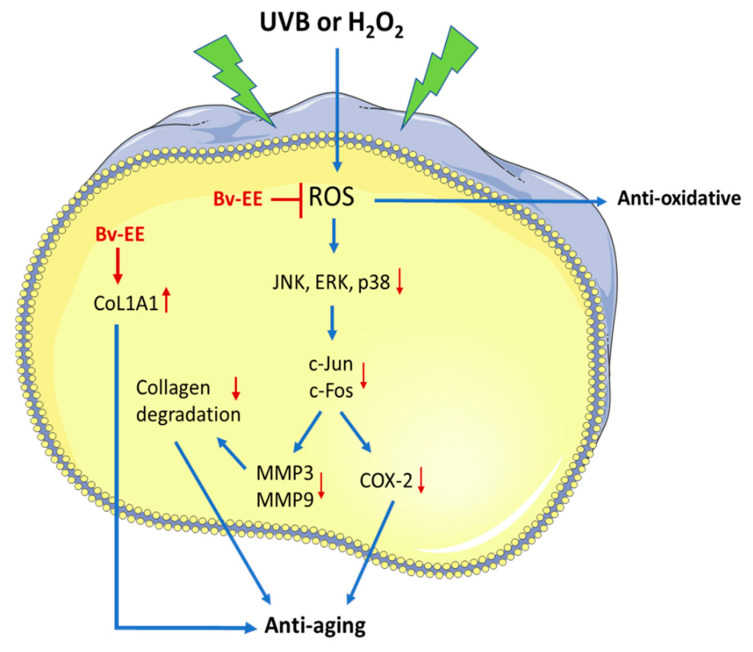
Schematic representation of skin protective effect of Bv-EE.

**Table 1 plants-12-01088-t001:** Phytochemical analysis of Bv-EE by GC-MS.

Peak NO.	RT	Name of the Compound	Peak Height	Corr. Area	Corr. % Max	% of Total
20	7.621	2,3-Dihydro-3,5-dihydroxy-6-methyl-4H-pyran-4-one	876,767	17,904,906	100.00%	13.46%
28	9.349	Lyxitol, 1-O-hexyl	149,742	13,972,799	78.04%	10.50%
29	9.492	3-(2-Hydroxyethyl)imidazole-2-thione	285,772	13,749,668	76.79%	10.34%
36	11.616	Carbamic acid	146,325	10,905,600	60.91%	8.20%
2	1.767	Acetic acid	251,106	8,966,954	50.08%	6.74%
26	8.985	1,2,3-Propanetriol 1-acetate	241,168	7,172,538	40.06%	5.39%
4	2.185	2-Propenoic acid	151,545	5,636,432	31.48%	4.24%
37	12.666	Benzoic acid, 4-hydroxy-	54,972	5,252,489	29.34%	3.95%
45	29.272	Hexestrol	111,097	4,721,710	26.37%	3.55%
40	13.965	Ethyl .beta.-d-riboside	99,327	3,979,699	22.23%	2.99%
30	9.606	1,8-Diamino-3,6-dioxaoctane	70,016	3,407,960	19.03%	2.56%
41	14.127	4-O-beta-D-galactopyranosyl	59,056	3,146,589	17.57%	2.37%
8	3.874	Butanoic acid	78,568	3,075,157	17.17%	2.31%
17	6.582	1,3,5-Triazine-2,4,6-	95,026	2,726,316	15.23%	2.05%
19	7.497	Ethanamine	52,324	2,486,491	13.89%	1.87%
21	7.803	Benzoic acid	34,833	2,246,052	12.54%	1.69%
44	23.777	2-(Acetoxymethyl)-3-(methoxycarbonyl)biphenylene	81,417	1,969,340	11.00%	1.48%
6	2.721	2-Propanone, 1-hydroxy-	90,883	1,732,730	9.68%	1.30%
9	4.031	Dihydroxyacetone	39,401	1,686,793	9.42%	1.27%
24	8.821	5-Hydroxymethylfurfural	37,226	1,395,927	7.80%	1.05%

**Table 2 plants-12-01088-t002:** Detailed conditions of the GC–MS analysis.

GC	Column	HP-5MS UI, 30 m × 250 μm × 0.25 μm
	Injection volume	1 μL of 100 mg/mL
	Inlet temperature	250 °C
	Injection mode	Split, 50:1
	Carrier gas	He, 1 mL/min
	Oven temperature	50 °C → heating rate 10 °C/min → 300 °C (hold 15 min)
	Aux heater temperature	300 °C
MS	Ionization mode	Electron ionization
	Detection temperature	230 °C
	Quadrupole temperature	150 °C
	Detection mode	Scan, m/z:33–150

**Table 3 plants-12-01088-t003:** Sequences of primers used in PCR.

PCR Type	Gene Name	Sequence (5′-3′)	
RT-PCR (human)	Col1A1	Forward	CAGGTACCATGACCGAGACG
		Reverse	AGCACCATCATTTCCACGAG
	GAPDH	Forward	GCACCGTCAAGGCTGAGAAC
		Reverse	ATGGTGGTGAAGACGCCAGT
Real-time PCR (human)	COX-2	Forward	CAGCATTGTAAAGTTGGTGGACTGT
		Reverse	GGGATTTTGGAACGTTGTGAA
	MMP3	Forward	TGTTAGGAGAAAGGACAGTGGTC
		Reverse	CGTCACCTCCAATCCAAGGA
	MMP9	Forward	GCCACTTGTCGGCGATAAGG
		Reverse	CACTGTCCACCCCTCAGAGC
	GAPDH	Forward	GCGCCCAATACGACCAAATC
		Reverse	GACAGTCAGCCGCATCTTCT

## Data Availability

The data used to support the findings of this study are available from the corresponding author upon request.

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
