# Peer review of "Anti-Oxidative and Anti-Aging Effects of Ethanol Extract of the Officinal Breynia (Breynia vitis-idaea) In Vitro"

_plants, 2023, doi:10.3390/plants12051088_

Round 1

Reviewer 1 Report

The paper merits the publication in the current form. 

Results support derived conclusion, references are relevant 

Author Response

  1. The paper merits the publication in the current form. Results support derived conclusion, references are relevant 

*: Thank you very much for your comment and approval of the paper.

Reviewer 2 Report

Abstract: the plant name has to be reported in italics.

References: please format according to journal guidelines.

Please correct typos in the text.

The authors analyzed the extract via GC-MS for a punctual description of the phytochemicals.

Additionally, they performed colorimetric analyses for determining phenolic compounds in the ethanol extract, where these compounds are expected to be present. However there are at least two other determinations that could be carried out for better defining the composition of the extract: colorimetric assay of total flavonoids and HPLC analysis of the phenolic/flavonoid compounds for determining the prominent phytochemicals at the basis of the scavenging/reducing properties of the extract.

Reviewer 3 Report

1. The introduction lacks information about the phytochemical composition of Breynia vitis-idaea and the dominant active compounds and their biological activity.

2.     The article should also include the English name of the plant in the title.

3.     How many samples were extracted in the phytochemical analysis study? Does Table 1 provide data for one extract only and how many measurements were carried out?

4.     It is recommended to describe the method used for determining the active compounds using GC-MS in detail in the Methods section. In the Materials section, the solvents and/or standards used in this analysis should be presented. The method for preparing the extract for analysis should be described. It is not clear at present which extract was used in the study, whether it was ethanol extract or the final product after lyophilization.

5.     It is recommended to differentiate the method for phytochemical composition analysis from the method for extract production.

6.     Why was a 3-hour extraction time chosen for extract production using ultrasonication? Did this have any impact on the stability of the active compounds? Were the extracts filtered before the concentration process? How was the concentration process by vacuum  of the extract controlled ? What was the appearance of the extract after the lyophilization process? A detailed description of the extract production method is recommended.

7.     The method for determining phenolic compounds does not provide literature sources used to develop the methodology.

8.     The conclusions should be shorter.

Author Response

Reviewer 3:

  1. The introduction lacks information about the phytochemical composition of Breynia vitis-idaea and the dominant active compounds and their biological activity.

***: Thank you very much for your comment. We have added information about anti-oxidative components contained in Bv-EE in introduction section [Line 48-50].

  1. The article should also include the English name of the plant in the title.

***: Thank you very much for your comment. We found this plant is known to be “officinal breynia” as an English name. So, we have changed the title according to this comment [see Line 3].

  1. How many samples were extracted in the phytochemical analysis study? Does Table 1 provide data for one extract only and how many measurements were carried out?

***: Thank you very much for your comment. We have analyzed GC-MS twice with the same extract and found almost same pattern. Although we did not prepare extracts according to different collection seasons, regions, and years, we are sure the GC-MS results regarding the patterns of included components. However, we agree with the intention of your comment. Since confirming the difference of ingredients in this extract is one of very important points in developing this extract as a cosmeceutical biomaterial. Therefore, we will continue this work mentioned above to complete standardization of this extract for industrial application. Relevant sentences with this point has been included in Line 86-90

  1. It is recommended to describe the method used for determining the active compounds using GC-MS in detail in the Methods section. In the Materials section, the solvents and/or standards used in this analysis should be presented. The method for preparing the extract for analysis should be described. It is not clear at present which extract was used in the study, whether it was ethanol extract or the final product after lyophilization.

***: Thank you very much for your comment. We have separated the method sections for extract preparation and GC-MS analysis. Also, we have supplemented both materials and method of GC-MS (Materials, Method 4.3) and listed the detailed information about the analysis (Table 2). In addition,1 mL of testing solution (100 mg/ml dissolved in methanol) of dried Bv-EE after removing both ethanol by evaporation and water by lyophilization was used to identify ingredients within the plant [Line 295-305 and Table 2].

  1. It is recommended to differentiate the method for phytochemical composition analysis from the method for extract production.

***: Thank you very much for your comment. We have divided the corresponding section into two independent methods (extract preparation: 4.2 and GC-MS: 4.3). [Line 286-293 and 295-305]

  1. Why was a 3-hour extraction time chosen for extract production using ultrasonication? Did this have any impact on the stability of the active compounds? Were the extracts filtered before the concentration process? How was the concentration process by vacuum of the extract controlled? What was the appearance of the extract after the lyophilization process? A detailed description of the extract production method is recommended.

***: Thanks very much for your comment. This is very good comment. 3-h extraction time was employed to maximize the extraction yield and to minimize degradation of phenolic components, according to our previous experience. Of course, we have filtered before the centration process as usual method. Vacuuming was not given any trouble when it was evaporated and lyophilized, fortunately. The extract was obtained in the form of completely dried conditions showing well-prepared powder with brown color. Therefore, we do not think preparation of Bv-EE gave any problem to affect its biological and pharmacological activities. Relevant additional sentences have been included in Line 290-293

  1. The method for determining phenolic compounds does not provide literature sources used to develop the methodology. 

***: Thank you very much for your comment. We have updated the references regarding determination of phenolic compounds in method section [Line 369].

  1. The conclusions should be shorter.

***: Thank you very much for your comment. We have edited the Conclusion section and made it shorter. So that the last sentence has been deleted [see L451-457].

Round 2

Reviewer 2 Report

The authors responded to most of the Reviewer's requests of correction.

Reviewer 3 Report

Dear Authors,

The study  is well designed.  The article is corrected. I have no comments.

Sincerely

Reviewer